# Anisotropic and Highly Sensitive Flexible Strain Sensors Based on Carbon Nanotubes and Iron Nanowires for Human–Computer Interaction Systems

**DOI:** 10.3390/ijms241713029

**Published:** 2023-08-22

**Authors:** Decheng Wu, Yinlei Su, Rui Li, Jingyuan Zhao, Li Yang, Pingan Yang

**Affiliations:** School of Automation, Chongqing University of Posts and Telecommunications, Chongqing 400065, China; wudc@cqupt.edu.cn (D.W.); s210302004@stu.cqupt.edu.cn (Y.S.); s220301074@stu.cqupt.edu.cn (J.Z.); 2020212705@cqupt.edu.cn (L.Y.)

**Keywords:** high-aspect-ratio V-structures, Fe NWs, CNTs, human–computer interaction

## Abstract

Flexible strain sensors for multi-directional strain detection are crucial in complicated hman–computer interaction (HCI) applications. However, enhancing the anisotropy and sensitivity of the sensors for multi-directional detection in a simple and effective method remains a significant issue. Therefore, this study proposes a flexible strain sensor with anisotropy and high sensitivity based on a high-aspect-ratio V-groove array and a hybrid conductive network of iron nanowires and carbon nanotubes (Fe NWs/CNTs). The sensor exhibits significant anisotropy, with a difference in strain detection sensitivity of up to 35.92 times between two mutually perpendicular directions. Furthermore, the dynamic performance of the sensor shows a good response rate, ranging from 223 ms to 333 ms. The sensor maintains stability and consistent performance even after undergoing 1000 testing cycles. Additionally, the constructed flexible strain sensor is tested using the remote control application of a trolley, demonstrating its high potential for usage in practical HCI systems. This research offers a significant competitive advantage in the development of flexible strain sensors in the field of HCI.

## 1. Introduction

In recent years, the rapid development of wearable, flexible devices has attracted the attention of many researchers [1,2,3,4]. In particular, the broad application of wearable, flexible devices revolutionizes traditional HCI systems [5,6]. While the input devices of traditional HCI systems optimize the user experience to a certain extent, users have little potential to access more cutting-edge forms of interaction [7]. Fortunately, this issue can be effectively solved by creating novel human–machine interfaces using wearable, flexible devices [8]. However, as the interaction between humans and machines involves a complex multi-directional strain scenario, it is essential to build flexible strain sensors with anisotropic features.

Heretofore, much effort has been made to improve the performance of flexible strain sensors. Flexible strain sensors achieve the detection of mechanical strain by relying on changes in the conductive network formed by the sensitive material, which in turn is caused by the deformation of the flexible substrate. Two key aspects of sensor performance are optimized, primarily in terms of sensitive material selection and structural design [9,10,11,12,13,14,15,16,17,18,19,20,21,22,23]. The main conductive filler materials widely used by researchers include carbon-based materials [24,25,26,27,28,29,30], metal-based materials [31,32,33,34], and electrically conductive polymers [35,36,37]. Carbon nanotubes have good electrical and mechanical properties and are widely used to fabricate high-performance sensors [38]. However, despite carbon’s excellent electrical conductivity, sensors prepared solely using carbon as the sensitive material are not sufficiently sensitive and reproducible. The sensor performance can be increased to a larger extent by using various conductive materials with various properties as filler materials for flexible strain sensors. Among them, the high-length and -diameter iron nanowires (Fe NWs) have a unique linear structure, significantly improving the conductive network structure and enhancing sensor sensitivity [39,40]. Therefore, using iron nanowire and carbon nanotube (Fe NWs/CNTs) mixtures as sensor filler materials considerably enhances sensor sensitivity and optimizes the conductive network structure of sensors.

To enhance the anisotropy of flexible strain sensors for strain detection in different directions, numerous researchers have introduced surface micromorphologies with orientation [41,42,43,44,45] or use oriented conductive materials as sensor-sensitive units [46,47,48,49,50]. For instance, Qiulin Wang et al. prepared an anisotropic flexible strain sensor by depositing carbon nanofibers (CNF)/polydimethylsiloxane (PDMS) onto the surface of an oriented thermoplastic polyurethane (TPU) nanofiber membrane [44]. Oriented carbon nanotubes and periodic folds were combined to prepare an anisotropic, extremely sensitive, flexible strain sensor by Heng Zhang et al. [46]. However, these periodic fold structures exhibit an uneven distribution of folds, which is a notable drawback. Furthermore, pleated structures created by pre-stretching the flexible substrate suffer from the disadvantage of low sensor sensitivity in the low-strain range. The incorporation of the sensitive material orientation adds complexity to the sensor fabrication process. Fortunately, the high-aspect-ratio V-groove array structures prepared via direct transfer exhibit a uniform distribution. Moreover, the high-aspect-ratio V-groove structure significantly alters the conductive network within the low-strain range, effectively enhancing sensor sensitivity. The V-groove structure, in particular, demonstrates considerably varied stretching effects when subjected to the same strain in different directions. When exposed to the same strain in different directions, such features allow the sensor’s conducting network to develop a large number of distinct cracks, causing the flexible strain sensor to exhibit substantial anisotropy. To develop highly sensitive and anisotropic flexible strain sensors for HCI applications, we consider introducing a high-aspect-ratio V-groove array structure and utilizing a hybrid Fe NWs/CNTs sensitive material. This combination optimizes the conductive network and greatly improves the sensor’s sensitivity across the entire detection range. Additionally, the sensor exhibits remarkable anisotropy in accurately detecting strains from various directions.

In this work, the researchers introduce a direct transfer method to prepare a flexible strain sensor substrate with a high-aspect-ratio V-groove array, and they spray Fe NWs and CNTs to form a hybrid conductive network, producing an anisotropic and highly sensitive flexible strain sensor. The PDMS substrate prepared via die transfer, instead of the conventional pre-stretching method, has a high-aspect-ratio V-shaped groove. On the one hand, it increases the sensitivity of the flexible strain sensor to the initial strain. The sensor exhibits a gauge factor (GF) of 2058.25. On the other hand, high-aspect-ratio V-groove arrays enable the sensor to exhibit exceptional anisotropy. In addition, the sensor demonstrates good stability through the ongoing 1000-cycle test. This highly sensitive anisotropic flexible strain sensor can be used as an input control device for HCI systems, enabling the remote control of vehicles. It also has the potential to be used as a wearable device for various applications in the field of HCI, guiding the design of other flexible strain sensors.

## 2. Results and Discussion

### 2.1. Characterization of V-Grooves and Conductive Networks

The surface morphology changes during the preparation of the sensor are tracked by SEM. The PDMS flexible sensor substrate with a high-aspect-ratio V-groove array configuration is obtained from the resin printed mold (Figure 1a,b). Additionally, a large number of stripes exist around each V-groove in the same direction as the V-groove growth (Figure 1c,d), and these fine stripes enable the sensitive material to adhere better to the surface of the flexible substrate. The direction of the stripes is consistent with the V-groove growth direction, which also enhances the anisotropy of the sensor.

The conductive network of Fe NWs is formed on the surface of the high-aspect-ratio V-groove array by spraying the Fe NWs solution with a mass fraction of 2% on the PDMS flexible substrate (Figure 1e,f). Additionally, in order to improve the sensor performance, researchers also sprayed the CNTs solution with a mass fraction of 1% to form a conductive network of Fe NWs and CNTs interleaved with each other (Figure 1g–i). Figure 1e–i demonstrate the feasibility of this approach.

The high and wide diameter of the sprayed sensitive material is more likely to accumulate at the bottom of the gully than the V-groove (Figure 1g). However, due to the narrow and deep design of the V-groove, even with more material accumulating at the bottom, it still maintains a distinct V-shaped surface profile. When the sensor is subjected to forces, the bottom of the high-aspect-ratio V-groove tends to deform more than the rest. The accumulation of sensitive material on the bottom fills in the conductive breaks that occur when the bottom is stressed. This property increases the sensor’s strain detection range to some extent [51,52,53]. (By using 5 mL of a 2% solution of iron nanowires and 5 mL of a 1% solution of CNTs to prepare the sensors, it is possible to reduce the buildup of sensitive material at the bottom of the V-grooves, at which point the sensors have a detection range of only 4%. However, the method described herein prepares a sensor with a detection range of 10%. (See Appendix A).)

The results above demonstrate that the PDMS flexible substrate prepared using a mold-transferred high-aspect-ratio V-groove array structure and then sprayed with a solution of Fe NWs and CNTs forms a hybrid conductive network with good conductivity. At the same time, the high-aspect-ratio V-grooves, with more sensitive material aggregating at the bottom, expand the strain detection range while maintaining high sensor sensitivity.

### 2.2. Tensile Sensing Performance of V-Groove Flexible Sensors

Flexible strain sensors with high-aspect-ratio V-groove arrays are characterized by high sensitivity, sensing anisotropy, and good reliability and can be easily packaged for use. The high-aspect-ratio V-groove array on the sensor surface achieves sensing anisotropy. There is a significant difference in sensitivity in different directions for the same strain. This feature effectively differentiates strains from different directions in practical applications. The Fe NWs/CNTs sensitive material coated on the surface of the high-aspect-ratio V-groove array forms a hybrid conductive network that changes during mechanical deformation of the sensor, causing a change in the resistance value. Since Fe NWs and CNTs have different mechanical properties, the use of Fe NWs/CNTs to build interleaved conductive networks effectively improves the sensitivity and operating range of the sensor [24,39].

The researchers had designed a series of tensile strain experiments to investigate the electromechanical properties of the sensor. The tensile strain experiments used sensors sprayed with 10 mL of Fe NWs solution with 2% Fe NWs by mass and 10 mL of CNTs solution with 1% CNTs by mass in a 10 m × 20 mm microstructure section. The effective area was 10 mm × 20 mm for the direction T stretching experiment (Figure 2a) and 10 mm × 14.43 mm for the direction L stretching experiment (Figure 2b). The resistance values were collected at a fixed sampling rate of 50 ms during the experiments. Figure 3a,c show the sensor performance for a 10% stretch in direction L and direction T, respectively, and the sensor sensitivity was evaluated using the general equation $\mathrm{GF}=(∆R/R_{0})/∆\varepsilon$ for GF, where ∆ε=∆L/L, ε denotes the strain applied to the sensor. The tensile sensing performance in directions T and L strongly demonstrated the very high sensitivity of sensors with high-aspect-ratio V-groove arrays (GF = 268.21 for 0–8% tensile strain, and GF = 2058.25 for 8–10% in direction L), and the significant difference in GF between the two directions also demonstrated the excellent anisotropy of sensors. The sensors showed good linearity in the different characteristic regions, with the linearity of the fit basically above 90%. The sensor’s good hysteresis and recoverability to direction L tensile strain is shown in Figure 3b,d. The comparatively weak hysteresis and recoverability for direction T was due to the contraction of direction L caused by stretching in direction T. Therefore, the distribution of sensitive material was somewhat influenced by the proximity of V-grooves to one another. However, the effect could have been negligible in practice since the sensor perceived the tensile strain in direction T as relatively insignificant.

In addition to the high sensitivity and anisotropic sensing characteristics, the sensor also had good stability and response speed. Figure 3e shows the sensor’s response time at 5% tensile strain and 5 mm/s tensile speed, with a response time of 333 ms when stretched and 223 ms when released. In order to demonstrate the good stability of the sensor, 1000 cycles were performed on the sensor in direction L and direction T at a tensile strain of 5% and a tensile velocity of 1 mm/s, respectively. Figure 3f shows the sensor repeatability experiment results. As the number of stretches increased, the sensor’s response in both directions gradually stabilized, and finally, the $\Delta R/R_{0}$ in direction L and direction T remained stable after 1000-cycle testing.

Figure 4 depicts the detection mechanism that allows the sensor to achieve anisotropy and high sensitivity. The high-aspect-ratio V-groove array structure achieves the anisotropy of the sensing function. The direction of the V-grooves and their arrangement is defined as direction T and direction L, respectively (Figure 4e). Since the strain in a certain direction can be decomposed into two orthogonal directions, it is only essential to analyze the sensing performance of the sensor in direction T and direction L.

When the tensile strain in the direction T occurs in the flexible strain sensor with a high-aspect-ratio V-groove array, it can be divided into three stages to analyze the strain process sensor characteristics: smaller stretch degree (Figure 4b), medium stretch degree (Figure 4c), and larger stretch degree (Figure 4d). Smaller stretching in direction T occurs, the sensor sensitive unit partially deforms, the hybrid conductive network composed of Fe NWs and CNTs changes, the gap between the sensitive materials of Fe NWs and CNTs in direction L increases, and part of the conductive network breaks, causing an increase in the resistance value (Figure 4h) [41,46]. A very slight contraction occurs in the direction L, but at this time, the conductive pathway disconnection caused by the larger spacing between the sensitive materials still plays a major role, and the resistance value increases continuously with the stretching (Figure 3c stage 0). As the stretching on the direction T further expands, the direction L shrinks. The spacing between the V-shaped grooves further decreases, and many Fe NWs/CNTs sensitive materials become closer and connected in the direction L. A large number of conductive networks are linked again (Figure 4a) [54], causing changes in the resistance value, and a decrease in the resistance value occurs with the increase in the tensile strain (Figure 3c stage 1). As the tensile strain in direction T increases beyond a certain critical value, the spacing between the V-shaped grooves can hardly be reduced further. The role of the sensitive material in causing the disconnection of the conductive network due to the larger spacing on the direction T is again dominant, and the increase in tensile strain further induces a change in the resistance value, which increases with the increase in tensile strain (Figure 3c, stage 2).

Flexible strain sensors with high-aspect-ratio V-groove arrays experience tensile strain in the direction L with a significant difference from the direction T (Figure 4f,g). There is no strain region where the resistance value decreases with increasing strain stretching. The reason is that although the direction L stretching causes contraction in the direction T, there is no V-shaped groove in the direction T, and the effect of spacing reduction between sensitive materials is not obvious. As the degree of strain on direction L keeps increasing, the resistance value keeps increasing [55,56]. However, because of V-grooves with a high aspect ratio in direction L, such a microstructure leads to a greater tensile strain in the deepest part of V-grooves when the sensor is stressed. When the degree of tensile strain in direction L is not large, although the V-groove spacing becomes wider, the sensitive material produces relative sliding, which effectively compensates for the disconnection of the conductive network pathway produced in the deepest part of the V-groove (Figure 4f). At this point, the conductive network disconnection caused by the larger spacing of the sensitive material on the sloping surface of the V-groove plays a dominant role (Figure 3a stage 0). However, as the tensile strain continues to increase, the gap between the tensile strain generated at the deepest part of the V-shaped gully and the tensile strain generated at the sloped portion of the V-shaped gully further widens. The sensitive materials’ relative sliding cannot make up for the difference in strain generated between them. A large number of conductive breaks are generated in the depths of the V-shaped gully back (the place marked by the red circle in Figure 4g), and an increase in sensitivity in the direction of the sensor, L, can cause this phenomenon (Figure 3a, stage 1).

In summary, the V-shaped trench array structure with a high aspect ratio brings remarkable anisotropy and high sensitivity to the sensor, and the conductive network composed of Fe NWs/CNTs fillers optimizes the sensitivity and stability of the sensor. Furthermore, the sensor’s simple preparation considerably boosts its economic benefits and has great competitive potential in the field of HCI.

### 2.3. Sensor Applications in HCI

A flexible sensor with high sensitivity in multiple directions can be prepared based on a single anisotropic high-sensitivity sensor, effectively detecting complex and weak human motions. This characteristic is used to prove the effectiveness of the sensor as an input device in HCI systems. It is worth noting that despite the successful realization of human motion detection using a single anisotropic sensor (as shown in Appendix A), the correlation between the strain degrees of our sensors in the T and L directions and the $\Delta R/R_{0}$ is not unique (as shown in Appendix A). Therefore, in order to more accurately detect input commands from human users in complex HCI systems, we propose a technique that stacks multiple independent anisotropic flexible sensors in a staggered configuration.

HCI utilized for the remote control of mobile equipment could significantly enhance the quality of life and is frequently used to replace people performing risky duties [36,57,58]. Therefore, it could be chosen as an experimental object for verifying the effectiveness of the prepared sensor in an HCI system. The sensor was worn on the wrist during the experiment, and the sensor resistance value was recorded in real-time using an A/D converter. The data was then transmitted to a computer for signal analysis and processing, parsing the user input commands. The cart’s motion was controlled wirelessly and remotely according to the commands (Figure 5a).

The experiments defined the two sensors’ V-groove array arrangement directions as the X and Y axes (Figure 5a). The downward pressure action and the left-to-right swinging action were two typical human motions with different directions, so the researchers analyzed the signals of these two types of human motions with typical characteristics during the experiments. The *X*-axis and *Y*-axis resistance value output signals of sensors differed when our wrist executed the downward pressing action and the lifting action (Figure 5b), which aided in accurately detecting these two activities. As demonstrated in Figure 5b, when we pressed down on our wrist, the *Y*-axis output resistance value sensor varied mainly in direction L, while the *X*-axis output resistance value varied mainly in direction T. The *Y*-axis output resistance value varied more drastically than the *X*-axis. Therefore, the value of the *Y*-axis output resistance value changed more strongly. In particular, the upper and lower sensors were squeezed relative to each other during the downward pressure process, and this feature caused a relative decrease in the resistance value (as shown in Appendix A). However, since the sensors had the characteristic of being more sensitive in direction L, the X-axis output value produced a relatively significant decrease.

The sensor displayed extremely different features from the downward pressure process when the researchers swung the wrist from side to side. As seen in Figure 5c, the resistance change rate increased in both axes as the wrist swung side to side because the upper and lower sensors were essentially not compressed. More importantly, during the left-to-right swing, the wrist deformation was quite tiny, and the sensors could accurately detect the smaller changes and had adequate anisotropy.

As shown in Figure 5b–d, it is clear that the sensors displayed considerable changes for various human motions, which was strong evidence for the outstanding anisotropy of the sensors we had constructed. Smaller variations in human motion could also be adequately detected and distinguished simultaneously. In Appendix A, the researchers remotely controlled the cart’s motion while wearing the ready, flexible sensor. This experimental finding unequivocally proved that the sensors designed could serve as input devices in the HCI system.

## 3. Materials and Methods

### 3.1. Materials

In this study, based on our previous studies, we successfully synthesized Fe NWs using a straightforward magnetic field assisted in situ reduction method. The Fe NWs have an average aspect ratio of 350 and a diameter of 60 nm [59,60]. Anhydrous ethanol (98% concentration) used is provided by Liaoning Aomai Trading Co. (Yingkou, China). Deionized water (18.25 MΩ) is obtained from an ultrapure water system (GYJ2-20 LS, Chongqing Huachuang). PDMS prepolymer and curing agent (Sylgard 184) are purchased from Dow Corning (Wiesbaden, Germany). CNTs (purity ≥ 90 wt. %, length < 10 μm, 10 nm < OD < 50 nm) are purchased from Time Nano Technology Co., Ltd. (Chengdu, China). The mold utilized for fabricating the flexible sensor substrate was created through 3D printing using a resin material. The mold featured V-shaped grooves that measured 250 μm in width and 500 μm in depth.

### 3.2. Methods

In this paper, a flexible strain sensor with high sensitivity and anisotropy is prepared via a sensitive material spraying method with a simple fabrication method and low economic cost. Firstly, the prepolymer and crosslinker are configured into PDMS solution according to 20:1 and poured into a PDMS mold (Figure 6a), which has side-by-side V-shaped grooves with a high aspect ratio; each groove is 250 μm wide and 500 μm deep; then, the PDMS solution is cured into a flexible film by drying at 75 °C for 6 h (Figure 6b), and the PDMS V-Groove flexible substrate layer is removed from the mold (Figure 6c). The prepared PDMS V-groove flexible substrate layer has a high-aspect-ratio V-groove array, which can be used directly as a sensing substrate. Next, a mixture of Fe NWs and anhydrous ethanol (Figure 6d, 2% mass ratio of iron nanowires 10 Ml) and CNTs and anhydrous ethanol (Figure 6e, 1% mass ratio of CNTs 10 Ml) was uniformly sprayed on the PDMS V-Groove flexible substrate layer, respectively. This way, only a small amount of conducting material is needed to form a large number of conducting networks on the surface of the V-groove array. Finally, a PDMS film overlaps the PDMS V-Groove flexible substrate layer to form the final sensor structure (Figure 6f). Finally, electrodes are attached to complete the package (Figure 6g).

### 3.3. Characterization

The sensor V-groove microstructure is observed using field emission scanning electron microscopy (thermo scientific Apreo 2C). Morphological observation and elemental analysis of the active part of the sensor after coating with Fe NWs and CNTs sensitive materials are conducted using a field emission scanning electron microscope (OXFORD ULTIM Max65). A tensile tester (Opto Sigma, TSD-601, Shenzhen, China) performs a cyclic tensile test, using a jig to hold both sensor ends and a resistive flexible strain transducer with an effective area of 10 mm × 20 mm. The maximum stretch is 10%, and the stretching speed is 1 mm/s or 5 mm/s. The resistance measuring instrument used in the tensile test is the source meter (34465A) from Keysight Technologies, Inc. (Santa Rosa, CA, USA).

## 4. Conclusions

In conclusion, we have prepared a flexible V-groove array anisotropic strain sensor with a high aspect ratio. A conductive network with excellent conductivity has been formed by spraying Fe NWs and CNTs solutions. The sensor preparation process has been simple, effective, and cost-effective. The sensors have been reasonably assembled into mutually orthogonal double-layer structures, and the anisotropy and high sensitivity of the sensors have been utilized to realize remote wireless operation of the trolley motion, which has proven the effectiveness of the sensors as an input device in complex HCI systems, as described below.

(1)The high-aspect-ratio V-groove has been designed to increase the sensor’s sensitivity and realize its anisotropy. This high-aspect-ratio V-groove design has enabled the sensor to achieve a 35.92-times-higher sensitivity variability in the mutually perpendicular directions L and T. Furthermore, the sensor has consistently maintained a high sensitivity in direction L across the entire operating range. It has exhibited a value of GF = 268.21 for stretching below 8% and a value of GF = 2058.25 for stretching between 8% and 10%. The sprayed Fe NWs solution and CNTs solution have formed a conductive network with excellent electrical conductivity, making the sensor preparation simple, with less-sensitive materials, and more economical.(2)The double-layer structure sensors are assembled in a mutually orthogonal way as the input devices of HCI system to realize the remote control of the motion of a trolley, and this has effectively proven that the sensor preparation method proposed in this paper can be applied in the complex HCI field.

While our current sensors have been successfully applied to complex human–computer interaction systems, they lack the ultra-high-pressure force-pixel detection sensitivity and fast sensor response demonstrated by VATJ sensors [61]. Additionally, they do not possess the impressive sub-pixel-level 3D haptic sensing capabilities observed in electronic skin embedded with IGZO sensors [62]. Nevertheless, it is worth highlighting that our sensors have anisotropic properties, unlike VATJ sensors. Furthermore, they effectively detect internal deformation by directly measuring the resistance value, which is a unique feature not found in electronic skins embedded with IGZO sensors. However, there are several areas requiring further research and improvement. These include addressing the shortcomings of insufficient response speed, limited detection range, and the inability to detect pressure pixels. In particular, it would be valuable to investigate the pressure detection capability of V-groove array-based sensors assembled into mutually orthogonal double-layer sensors. Overall, our sensors demonstrate certain advantages while acknowledging areas for improvement. Further exploration and development will contribute to their enhanced performance and expanded functionalities.

## Figures and Tables

**Figure 1 ijms-24-13029-f001:**
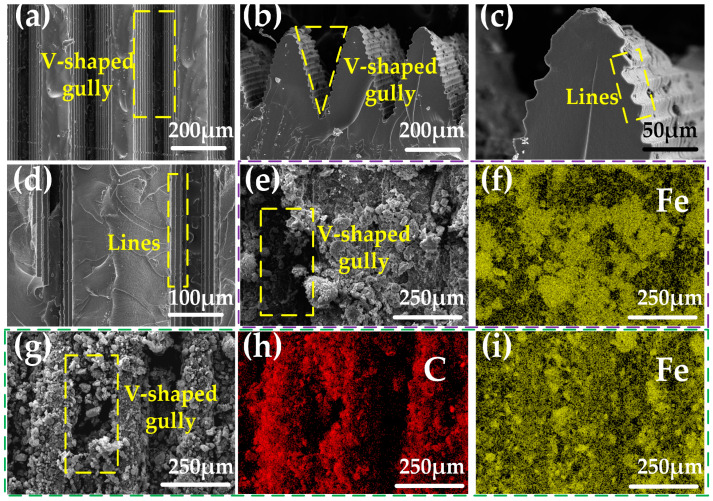
Scanning electron microscope (SEM) image of the V-groove sensor. (**a**,**d**) Surface electron micrograph of PDMS flexible base. (**b**,**c**) Electron micrograph of PDMS flexible base cross-section. (**e**,**f**) Electron microscope image of the sensor surface after spraying Fe NWs. (**g**–**i**) Electron micrographs of the sensor surface after spraying Fe NWs and CNTs.

**Figure 2 ijms-24-13029-f002:**
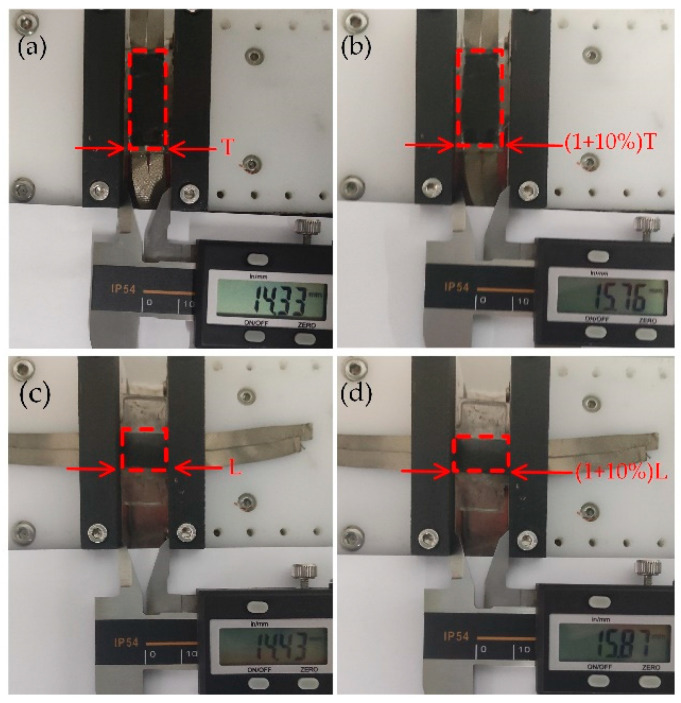
(**a**) T-direction original state. (**b**) T-direction stretching (by 10%). (**c**) L-direction original state. (**d**) L-direction stretching (by 10%).

**Figure 3 ijms-24-13029-f003:**
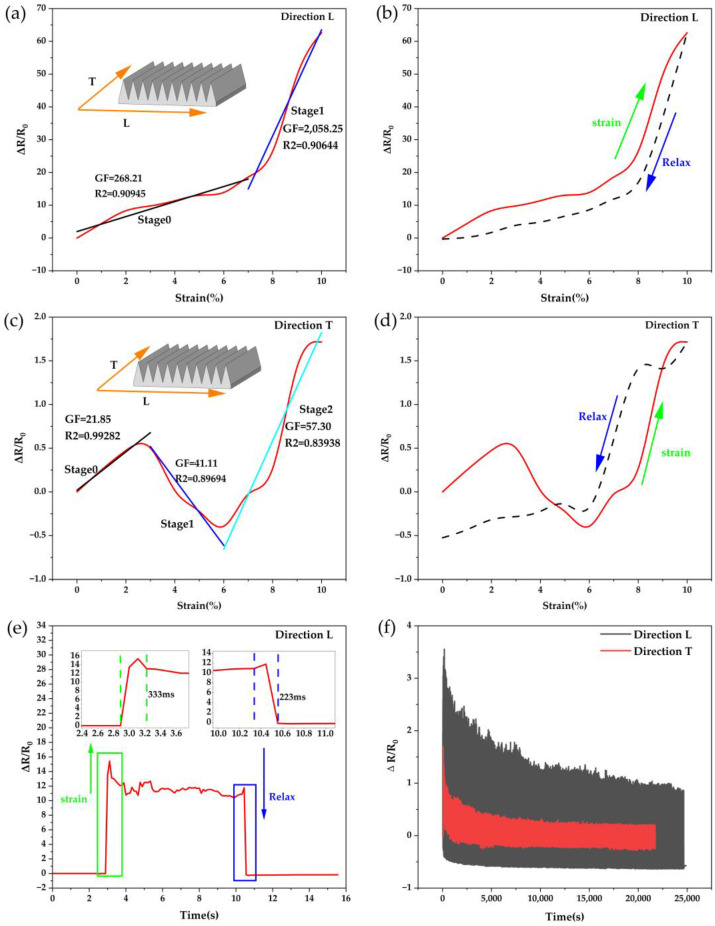
(**a**) Gauging factor and linearity of the sensor stretched in the L direction. (**b**) Sensor hysteresis in the L direction. (**c**) The gauging factor and linearity of the sensor stretched in the T direction. (**d**) Sensor hysteresis in the T direction. (**e**) Sensor response time of the sensor in the L direction (stretched by 5%). (**f**) The sensor stretched by 5% in the L and T directions, repeated for 1000 cycles.

**Figure 4 ijms-24-13029-f004:**
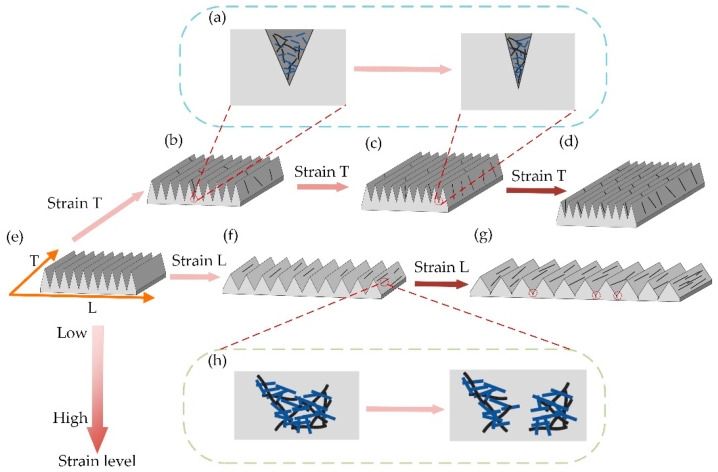
(**a**) V-groove contraction conductive path change. (**b**–**d**) Flexible strain transducer in direction T stretch conductive path change. (**e**) Flexible strain transducer initial state. (**f**,**g**) Flexible strain transducer in direction L stretch conductive path change. (**h**) Flexible strain transducer stretch generation.

**Figure 5 ijms-24-13029-f005:**
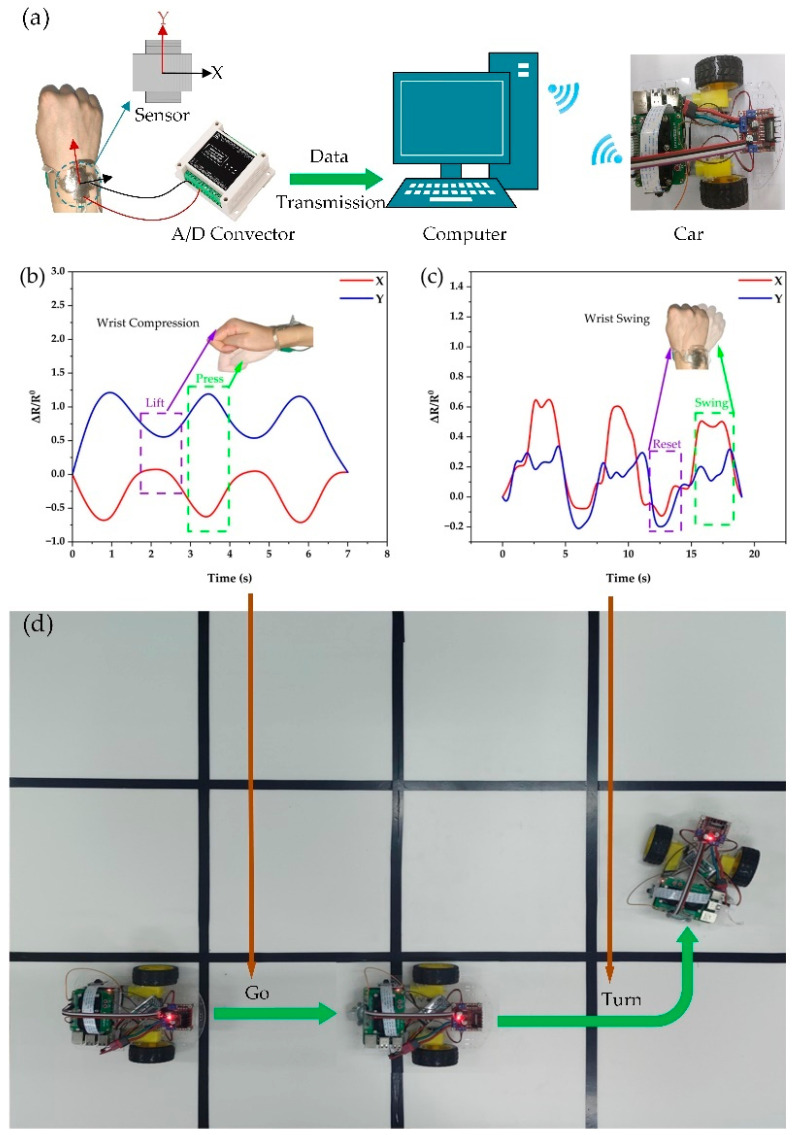
Demonstration of anisotropic sensors as HCI devices to control the cart. (**a**) HCI system diagram. (**b**) Signals collected by the sensors when the wrist is pressed. (**c**) Signals collected by the wrist swing sensors. (**d**) The cart moves according to the control command.

**Figure 6 ijms-24-13029-f006:**
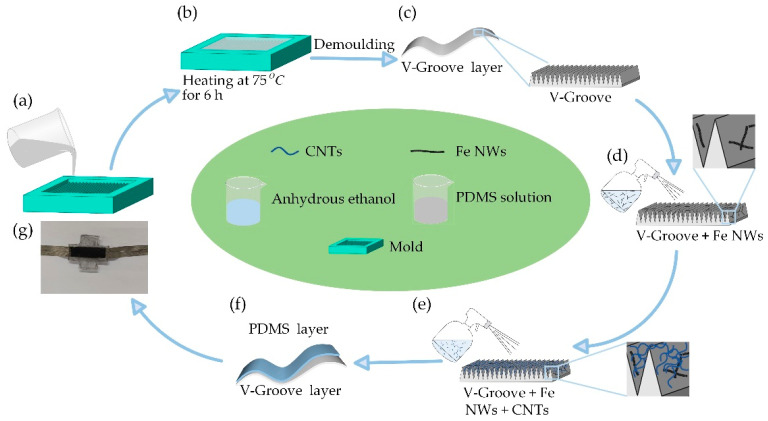
Sensor preparation flow. (**a**) PDMS solution with 20:1 A:B components is poured into the mold. (**b**) Drying at 75 °C for 6 h. (**c**) Demold to obtain PDMS V-Groove flexible base layer. (**d**) Spray a 2%-by-mass mixture of iron nanowires and anhydrous ethanol on the PDMS flexible base. (**e**) Spray a mixture of carbon nanotubes and anhydrous ethanol with a mass ratio of 1% on the PDMS flexible base. (**f**) V-Groove sensor encapsulation. (**g**) Encapsulation of the obtained sensor.

## Data Availability

Not applicable.

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
