# Peer review of "Anisotropic and Highly Sensitive Flexible Strain Sensors Based on Carbon Nanotubes and Iron Nanowires for Human–Computer Interaction Systems"

_ijms, 2023, doi:10.3390/ijms241713029_

Round 1

Reviewer 1 Report

In this manuscript, the Authors describe a method for creating a flexible strain sensor substrate with a high aspect ratio V-groove array. They used Fe NWs and CNTs to spray coat an interlocking conductive network, resulting in an anisotropic flexible strain sensor. The sensor demonstrated a gauge factor of approximately 2,000, 2-dimensional anisotropy, and significant stability after 1000 stretching cycles. Furthermore, the sensor was used to remotely control a small vehicle through the HCI system. My recommendation is a revision to address some minor and major corrections.

1. This Reviewer is wondering about the possible causes that seem to limit the sensor response time. How to explain the response times in the range of a hundred milliseconds? Perhaps this range is a consequence of the mechanical stimulus time or the resistance acquisition time.

2. When considering active-matrix applications like multiplex HCI and artificial skin, it's important that the strain sensor response is able to detect anisotropic pressure. Can the authors provide an estimate of (and discuss) the pressure sensitivity (in Pa^-1) along both L and T directions?

3. Please compare their sensor performance to magnetic (doi.org/10.1038/s41467-022-29802-7) and variable-area (https://doi.org/10.1021/acsami.8b12212) strain sensors.

4. Considering scalability, performance, and operational range, can V-groove array technology enhance, integrate, or provide additional features to the aforementioned pressure-pixel technologies?

5. The demonstration of anisotropic sensors as HCI devices is insightful. However, as the L-direction GF is up to 2 magnitude orders higher than the T-direction GF, using two L-direction sensors connected to each other (90-degree rotated) may result in more precise control of the small car. Can the authors reinforce the advantages of using a single anisotropic sensor even with a potentially less precise control?

A minor to moderate editing of the English is required to improve the language style. It is worth mentioning the current form does not compromise the quality of the discussion.

Author Response

Thank you very much for your careful review and constructive suggestions with regard to our manuscript. The insightful comments provided have greatly contributed to the revision and enhancement of our paper, serving as a crucial guide in our research endeavor. We have conducted a thorough examination of the manuscript and exerted our utmost efforts to revise and improve it significantly, taking into account the favorable comments received from the reviewers. The revised sections have been appropriately highlighted in red within the paper. Notably, substantial corrections have been made throughout the article, aligning with the reviewers' comments, and our responses have been rendered in a coherent manner. Our heartfelt appreciation extends to the editor and reviewers for their enthusiastic dedication to this work. We sincerely hope that the revisions will meet their approval and result in a successful outcome.

Reviewer 2 Report

The authors enhanced the anisotropy and sensitivity of the sensors for multi-directional detection using a simple and effective method. In their approach, they proposed a flexible strain sensor with anisotropy and high sensitivity based on a high aspect ratio V-groove array and a hybrid conductive network of Fe NWs/CNTs. The authors claim that their developed sensor shows significant anisotropy with a deference strain detection sensitivity upto 35.92 in the two mutually perpendicular directions. Furthermore, their sensor shows a good dynamic performance response rate of 223 to 333 ms, showing stability and consistence performance for 1000 cycles. Finally, they demonstrated its high potential usage in HCI systems, testing it with a remoted controlled trolley.

Overall, the paper is well written, explanations and demonstrations are good and is comprehensive. I particularly like the demonstration part in the video showing control along T and L directions. In general, I find it interesting.

However, I would suggest acceptance after modifications/improvement of the below comments.

Specific comments are mentioned below:

1.     In the abstract and line 44, the authors should define the full form of the abbreviation Fe NWs and Fe NW/CNTs. It would be easier to understand for a general audience.

2.     There is some unnecessary extra space between words throughout the manuscript. For e.g, in line 12, 28, 34 etc, the authors can modify that.

3.     In Fig 2, the texts on the images are difficult to read. At least I was not able to read it in Fig 2 (a,b,c,d,e,g). The authors should make the texts clearer.

4.     In lines 153, 154, and 155, the authors say, ‘The accumulation of sensitive material on the bottom fills in the conductive breaks that occur when the bottom is stressed. This property increases the sensor's strain detection range to some extent.’ Is there any proof or reference for this behavior? The authors can add a citation if there is a supporting work or illustrate this point if no work is available yet. 

5.     The following sentence in lines 157-159, ‘The results above demonstrate that the PDMS flexible substrate prepared using a mold-transferred high aspect ratio V-groove array structure and then sprayed with a solution of Fe NWs and CNTs forms an interlocking conductive network with good conductivity.’ seems a bit vague. I do not see any evidence of an interlocking conductive network or any measured conductivity values. The authors should provide more proof or explain this.

6.     Does the anisotropy of CNTs morphology play any role? Do the authors have any SEM or explanation of what happens to the CNTs during the T and L directions?

7.     Any brief indication of shortcomings in the study and comment on future works in the conclusion part would be helpful for the readers.

Author Response

(The authors gave the same response as above.)

Round 2

Reviewer 1 Report

After carefully considering the comments and suggestions provided by the Reviewer, the Authors have made improvements to the manuscript. Therefore, I recommend that the manuscript be accepted in its current form.

A minor editing of the English is required.

Reviewer 2 Report

I'm satisfied with the authors addressing the comments.